# The Effects of Immunosuppressive Drugs on the Characteristics and Functional Properties of Bone Marrow-Derived Stem Cells Isolated from Patients with Diabetes Mellitus and Peripheral Arterial Disease

**DOI:** 10.3390/biomedicines11071872

**Published:** 2023-06-30

**Authors:** Jitka Husakova, Barbora Echalar, Jan Kossl, Katerina Palacka, Vladimira Fejfarova, Michal Dubsky

**Affiliations:** 1Diabetes Centre, Institute for Clinical and Experimental Medicine, 14021 Prague, Czech Republic; jitkahazdrova@seznam.cz (J.H.); vlfe@ikem.cz (V.F.); 2First Faculty of Medicine, Charles University, 14021 Prague, Czech Republic; 3Department of Nanotoxicology and Molecular Epidemiology, Institute of Experimental Medicine of the Czech Academy of Sciences, 14220 Prague, Czech Republic; barbora.echalar@gmail.com (B.E.); kossl.jan@gmail.com (J.K.); katerina.palacka@iem.cas.cz (K.P.); 4Department of Cell Biology, Faculty of Science, Charles University, 14021 Prague, Czech Republic

**Keywords:** cell-based therapies, diabetes, diabetic foot ulcers, immunosuppressive drugs

## Abstract

Background: Diabetic patients (DPs) with foot ulcers can receive autologous cell therapy (ACT) as a last therapeutic option. Even DPs who have undergone organ transplantation and are using immunosuppressive (IS) drugs can be treated by ACT. The aim of our study was to analyze the effects of IS drugs on the characteristics of bone marrow-derived stem cells (BM-MSCs). Methods: The cells were isolated from the bone marrow of DPs, cultivated for 14–18 days, and phenotypically characterized using flow cytometry. These precursor cells were cultured in the presence of various IS drugs. The impact of IS drugs on metabolic activity was measured using a WST-1 assay, and the expression of genes for immunoregulatory molecules was detected through RT-PCR. Cell death was analyzed through the use of flow cytometry, and the production of cytokines was determined by ELISA. Results: The mononuclear fraction of cultured cells contained mesenchymal stem cells (CD45^−^CD73^+^CD90^+^CD105^+^), myeloid angiogenic cells (CD45^+^CD146^−^), and endothelial colony-forming cells (CD45^−^CD146^+^). IS drugs inhibited metabolic activity, the expression of genes for immunoregulatory molecules, the production of cytokines, and the viability of the cells. Conclusions: The results indicate that IS drugs in a dose-dependent manner had a negative impact on the properties of BM-MSCs used to treat ischemic diabetic foot ulcers, and that these drugs could affect the therapeutic potential of BM-MSCs.

## 1. Introduction

Diabetes mellitus (DM) is a chronic metabolic disease associated with high morbidity and mortality, and its incidence has significantly increased in the last decade [1]. The most common manifestation of this disease is hyperglycemia, which is due to the absolute or relative lack of insulin or the inability of cells to respond to insulin production. DM is divided into two of the most common types. DM type 1 is characterized by a defect in the secretion of insulin associated with the immunologically mediated destruction of β-cells in the pancreas. In DM type 2, there is a specific progressive resistance to insulin, especially in the liver, muscles, and fat tissue. In addition, there is gestational diabetes as well as other specific types of DM, which may be caused by a genetic defect in β-cells or insulin production, glucotolerance changes, etc. [2]. If DM is not properly controlled, it can lead to several chronic complications such as diabetic kidney disease, retinopathy, neuropathy, macroangiopathy, or diabetic foot ulcers (DFUs) [3,4,5].

For patients with DFUs who cannot be treated with percutaneous transluminal angioplasty or other standard methods, autologous cell therapy (ACT) serves as a last resort treatment option [6,7,8]. Stem cells (SCs), with their ability to self-renew and differentiate into many cell types, show great potential for use in medical therapies. The main benefit of SC therapy in treating DFUs is the induction of therapeutic angiogenesis and neovascularization, which leads to an increase in blood flow and tissue regeneration [9,10]. The most common source of SCs for ACT is bone marrow (BM). BM-derived mesenchymal stem cells (BM-MSCs) may contribute to the healing process through their ability to produce various cytokines and growth factors as well as to differentiate into diverse cell lineages [11,12]. MSCs are characterized by the expression of cell surface markers CD90, CD73, and CD105 and the lack of markers CD45, CD11b, and CD34. MSCs must be able to differentiate into osteocytes, chondrocytes, and adipocytes in vitro and to adhere to plastic surfaces in standard culture conditions [13]. Other populations of BM-SCs include non-adherent vascular precursors of endothelial colony-forming cells (ECFCs) and myeloid angiogenic cells (MACs), which also contribute to the revascularization process. MACs are defined by the expression of cell markers CD45, CD14, and CD31 and the lack of CD146 and CD34. ECFCs, on the other hand, are CD146^+^ CD45^−^, and CD14^−^ [14,15].

Patients with DFUs who have undergone solid-organ transplantation and who take immunosuppressive (IS) drugs can also be treated with ACT. The most commonly prescribed IS drugs following kidney or pancreas transplantations are tacrolimus, sirolimus, and mycophenolate mofetil, all of which have different mechanisms of action: sirolimus is a mechanistic target of rapamycin inhibitors; mycophenolate mofetil inhibits inosine monophosphate dehydrogenase; and tacrolimus is an inhibitor of calcineurin. However, even though these drugs provide positive therapeutic effects, they come with several side effects that include altering the function, differentiation, and immunomodulatory properties of precursor cells [16,17]. Studies have shown that IS drugs inhibit the viability and proliferation of different cells, cause gut microbiota dysbiosis and urinary tract disorders, and have many other negative effects on cell populations [18,19,20]. Moreover, IS drugs could negatively impact SCs, thereby inhibiting the therapeutic effect of ACT. The majority of studies that have examined the influence of IS drugs on precursor cells were performed on cells obtained from non-diabetic patients. However, it has been shown that when compared to control vascular precursors obtained from non-diabetic patients, vascular precursors isolated from DPs are less functionally potent and have lower expression of phenotypical markers [21]. IS drugs could therefore have a different impact on cells isolated from DPs. For this reason, the aim of our study was to evaluate the effect of IS drugs on the characteristics and functional properties of BM-SCs obtained from DPs. We revealed that IS drugs decreased both the metabolic activity and gene expression for immunoregulatory molecules and inhibited the production of cytokines and growth factors by BM-SCs isolated from patients with DM. In addition, the rate of apoptosis and necrosis of these SCs increased in the presence of IS drugs.

## 2. Materials and Methods

### 2.1. Collection of BM

At the Institute for Clinical and Experimental Medicine, a mononuclear fraction of cells (290 mL) was collected from the BM of 6 DPs aged 50–84 years during ACT under aseptic conditions. The donors were patients with DM and non-healing DFU, chronic limb-threatening ischemia, and transcutaneous oxygen pressure below 30 mm Hg. They had not been diagnosed with cancer or any other severe diseases in the previous 5 years, had not suffered a myocardial infarction or stroke in the previous 6 months, and had a life expectancy of at least 6 months. They had no signs of neoplastic process or unstable proliferative retinopathy and were able to undergo local or general anesthesia. All patients provided written informed consent for the use of BM fraction for experimental purposes, and this study was approved by the Ethics Committee with multi-center competence of the Institute for Clinical and Experimental Medicine and Thomayer Hospital. A total of 250 mL of bone marrow-derived cells was used for ACT, with the rest being used for this study.

### 2.2. Isolation and Cultivation of BM-SCs

The suspension of BM was separated with gelofusin (B Braun Melsungen AG, Melsungen, Germany), and the erythrocyte-free fraction of cells was seeded in 15 mL of Dulbecco’s Modified Eagle Medium (DMEM, Sigma-Aldrich, St. Louis, MI, USA) containing 10% fetal calf serum (FCS), antibiotics (100 µg/mL of streptomycin, 100 U/mL of penicillin), and 10 mM HEPES buffer (complete DMEM). We only used directly harvested cells and did not freeze or store any cell populations for future experiments. The cells were cultured in a final concentration of 5 × 10^6^ cells/mL in 75-cm^2^ tissue culture flasks (Techno Plastic Products (TPP), Trasadingen, Switzerland). Adherent and non-adherent cells were cultured for 14–18 days at 37 °C with a regular exchange of medium and passaging in order to maintain an optimal number of cells. The cells were harvested in the 3rd to 4th passage through incubation with 1 mL of a 0.5% trypsin solution (Sigma-Aldrich) for 5 min followed by a gentle scraping. After 14–18 days of cultivation, the suspension of adherent and non-adherent cells was used for further experiments.

### 2.3. Phenotype Characterization of BM-SCs by Flow Cytometry

BM-SCs were washed in phosphate-buffered saline and incubated for 30 min (4 °C) with the monoclonal antibodies allophycocyanin APC-labeled anti-CD146 (clone P1H12, BioLegend, San Diego, CA, USA), phycoerythrin PE-labeled anti-CD105 (clone SN6, Life Technologies, Carlsbad, CA, USA), fluorescein isothiocyanate FITC-labeled anti-CD45 (clone HI30, BioLegend), FITC-labeled anti-CD90 (clone 5E10, BioLegend), PE-labeled anti-CD73 (clone AD2, eBioscience, San Diego, CA, USA), and APC-labeled anti-CD45 (clone HI30, BioLegend). Dead cells were stained using Hoechst 33,258 fluorescent dye (Invitrogen, Carlsbad, CA, USA) and added to the samples 10 min prior to flow cytometry. Data (100,000 events per sample) were obtained using a LSRII flow cytometer (BD Biosciences, Franklin Lakes, NJ, USA) and analyzed using FlowJo 10 software (Tree Star, Ashland, OR, USA).

### 2.4. Measuring Metabolic Activity of BM-SCs in the Presence of IS Drugs

BM-SCs in a final concentration of 25 × 10^3^ cells/mL were cultured in a volume of 200 µL of complete DMEM in 96-well tissue culture plates (Nunc, Roskilde, Denmark) for 48 h untreated or in the presence of tacrolimus, sirolimus, and mycophenolate mofetil (purchased from Sigma-Aldrich) in a concentration ranging from 0.005–500 µg/mL. The metabolic activity of the BM-SCs was determined using a WST-1 assay (Roche, Mannheim, Germany) and a Tecan Sunrise spectrophotometer (Life Science, Mannedorf, Switzerland), and analyzed by Kim 32 software (Schoeller Instruments, Prague, Czech Republic). The WST-1 solution (10 µL/100 µL of the sample) was added to samples for the final 3 h of the 48 h incubation period. The WST-1 is tetrazolium salt, which is cleaved to formazan through a cell mechanism, localized on a cell surface dependent on the production of NAD(P)H. Therefore, the formation of formazan correlates with the number of metabolically active cells.

### 2.5. Determination of the Impact of IS Drugs on the Expression of Genes of Immunoregulatory Molecules

BM-SCs were cultured in a final concentration of 8 × 10^4^ cells/mL in 24-well tissue culture plates (TPP) in a volume of 1 mL of complete DMEM, either unstimulated or stimulated with proinflammatory cytokines human recombinant IFN-γ, a final concentration of 10 ng/mL (PeproTech, Rocky Hill, NJ, USA), and human recombinant TNF-α (10 ng/mL) (PeproTech), or stimulated with cytokines in the presence of IS drugs (tacrolimus, sirolimus, mycophenolate mofetil) in a concentration ranging from 0.005–500 µg/mL for 48 h. The expression of genes for indoleamine 2,3-dioxygenase (IDO), cyclooxygenase-2 (COX2), programmed death-ligand 1 (PD-L1), an inducible isoform of nitric oxide synthase (iNOS), and transforming growth factor-β (TGF-β) (all primers from Generi Biotech, Hradec Kralove, Czech Republic—the sequences of primers are shown in Table 1) were determined by real-time PCR (RT-PCR). In brief, the total RNA was isolated from samples collected in TRI Reagent (Sigma-Aldrich). One µg of the total RNA was mixed with deoxyribonuclease I (Promega, Madison, WI, USA) and used for reverse transcription. The RT-PCR was processed using Power SYBR Green PCR master MIX (Applied Biosystems, Carlsbad, CA, USA) on cycler StepOne Plus RT-PCR System (Applied Biosystems). The RT-PCR parameters comprised denaturation at 95 °C for 3 min, 40 cycles at 95 °C for 20 s, annealing at 60 °C for 30 s, and elongation at 72 °C for 30 s. Data were collected at each cycle after an elongation at 80 °C for 5 s and were analyzed using StepOne Software 2.2.2 (Applied Biosystems, Waltham, MA, USA).

### 2.6. Evaluation of the Effects of IS Drugs on the Production of Cytokines and Growth Factors

BM-SCs were cultured using the same method implemented to determine the impact of IS drugs on the expression of genes of immunoregulatory molecules. The supernatants were collected after 48 h of cultivating untreated BM-SCs as well as those exposed to IS drugs. The production of hepatocyte growth factor (HGF), vascular endothelial growth factor (VEGF), interleukin 8 (IL-8), IL-6, and chemokine C-C motif ligand 2 (CCL2) were determined by ELISA kits (R&D Systems, Minneapolis, MN, USA).

### 2.7. The Impact of IS Drugs on Cell Death

To determine the effects of IS drugs on cell death, BM-SCs were cultured in a final concentration of 8 × 10^4^ cells/mL in 1 mL of complete DMEM in 24-well tissue culture plates (TPP). They were either untreated or exposed to sirolimus, tacrolimus, or mycophenolate mofetil in concentrations ranging from 0.005–500 µg/mL for 48 h. The percentage of apoptotic cells was determined through flow cytometry using an Annexin V apoptosis detection kit (Exbio, Prague, Czech Republic) in accordance with the manufacturer’s protocol. Annexin V binds to phosphatidylserine (a marker of apoptotic cells) when it appears in the outer layer of the plasma membrane. Dead cells were stained with Hoechst 33,258 fluorescent dye (Sigma-Aldrich) and added to the samples 10 min prior to flow cytometry. Data were collected using a LSRII cytometer (BD Biosciences) and analyzed by FlowJo 10 software (Tree Star).

### 2.8. Statistical Analysis

The results are expressed as the mean + standard deviation (SD). BM-SCs isolated from 6 patients were used for experimental purposes. Differences between the two groups were analyzed using the unpaired Student’s t-test and GraphPad Prism 5 (GraphPad Software, San Diego, CA, USA). A value of *p* < 0.05 was considered statistically significant.

## 3. Results

### 3.1. The Characterization of BM-SCs

BM-SCs (3rd–4th passage) were characterized using flow cytometry. Analyses of the expression of cell surface markers were performed on a population of live single cells. Cells characterized as CD45^−^CD90^+^CD73^+^ and CD105^+^ were considered MSCs; CD45^+^ and CD146^-^ were determined as MACs; and CD45^−^ and CD146^+^ were defined as ECFCs. Figure 1A illustrates a gating strategy of adherent and nonadherent BM-SCs. MACs represented approximately 23% of CD45^+^ cells, ECFCs represented 1.3% of CD45^−^ cells, and MSCs accounted for more than 90% of CD45^−^ BM-SCs (Figure 1B).

### 3.2. The Effects of IS Drugs on Metabolic Activity

To determine the effects of IS drugs on metabolic activity, BM-SCs were cultured in the presence of various IS drugs in different concentrations for 48 h, and the metabolic activity of these cells was determined through a WST-1 assay. Figure 2 shows that all IS drugs significantly decreased the metabolic activity of BM-SCs in concentrations of 50 µg/mL and higher.

### 3.3. The Impact of IS Drugs on the Expression of Genes of Immunoregulatory Molecules

The cells were cultured in 3 different ways: (1) unstimulated, (2) stimulated with IFN-γ and TNF-α, or (3) stimulated with cytokines in the presence of sirolimus, tacrolimus, or mycophenolate mofetil in concentrations of 0.005–500 µg/mL. After 48 h of cultivation, BM-SCs were harvested and then placed into TRI reagent, and the expression of genes for COX2, IDO, PD-L1, TGF-β, and iNOS was determined by RT-PCR. IS drugs altered the expression of genes of immunoregulatory molecules. Tacrolimus inhibited the expression of genes for COX2, IDO, PD-L1, TGF-β, and iNOS in a dose-dependent manner. Sirolimus had the most significant inhibitory effect on the expression of genes for iNOS. Conversely, the expression of the genes for iNOS increased in the presence of mycophenolate mofetil (Figure 3).

### 3.4. The Effects of IS Drugs on the Production of Cytokines and Growth Factors

BM-SCs were cultured in three different ways: (1) unstimulated, (2) stimulated with IFN-γ and TNF-α, or (3) stimulated with cytokines in the presence of different concentrations of IS drugs. The production of cytokines HGF, VEGF, CCL2, IL-8, and IL-6 was determined by ELISA after 48h of cultivation. As shown in Figure 4, IS drugs significantly decreased the production of all of these cytokines and growth factors. The production of HGF was significantly inhibited in the presence of sirolimus, mycophenolate mofetil, and tacrolimus in concentrations of 5 µg/mL and higher. Tacrolimus proved to be the most potent inhibitor of CCL2, IL-8, and IL-6 production. Sirolimus brought about the most pronounced decrease in VEGF production (Figure 4).

### 3.5. The Impact of IS Drugs on the Cell Death of BM-SCs

IS drugs decreased the metabolic activity and other functional properties of BM-SCs. We determined the percentage of apoptotic and dead BM-SCs in the presence of tacrolimus, sirolimus, and mycophenolate mofetil. Figure 5A shows the common gating strategy of apoptotic and dead cells. Apoptosis (Figure 5B) and the number of dead BM-SCs (Figure 5C) increased after exposure to IS drugs. Sirolimus gave rise to the highest number of apoptotic cells (Figure 5B), whereas tacrolimus proved to be the most potent enhancer of cell death (Figure 5C).

## 4. Discussion

Tacrolimus, sirolimus, and mycophenolate mofetil are among the most common IS drugs used by clinics to prevent and suppress rejection after solid-organ transplantation. However, IS drugs also come with various side effects, including an inhibitory effect on precursor cells and the healing process. IS drugs interfere with mesenchymal stem cell function and inhibit their activity [17]. It has been shown that tacrolimus negatively influences the viability, differentiation potential, and morphology of cells derived from the gingiva [18]. Another IS drug, cyclosporine A, inhibits the proliferation and adipogenous differentiation of SCs and alters their immunoregulatory properties [16]. Stem cell apoptosis is also enhanced when they are exposed to cyclosporine A. These findings suggest that IS therapy in combination with SC-based treatment could negatively affect the beneficial properties of vascular precursors in tissue healing.

It has been proven that sirolimus suppressed the lymphocyte proliferation and decreased the interleukin 17A secretion in patients after kidney transplantation [22]. Vitiallo et al. proved that mammalian targets of rapamycin inhibitors (siromimus and everolimus) showed a very potent anti-inflammatory effect by decreasing the response of neutrophils and decreasing both release of VEGF and IL-8 [23]. Moreover, Nowak et al. demonstrated the pro-apoptotic effect of sirolimus induced by oxidized LDL as well as mycophenolate mofetil—both drugs significantly increased the cell death in a monocyte cell lineage [24]. To our knowledge, there are no trials assessing the impact of IS drugs on the functional properties of hematopoietic stem cells; most of the studies are focused on the transplantation of these cells in the treatment of hematological diseases.

IS therapy is also provided to DPs after solid-organ transplantation. These patients often suffer from diabetic complications such as DFUs [5,25,26]. ACT is a last-resort clinical treatment option for DPS with DFUs [27,28]. The therapeutic potential of BM-SCs has been well-documented [29,30,31,32]. These cells contribute to tissue healing through their immunomodulatory abilities and by the production of various cytokines and growth factors [32]. One mechanism that is essential for tissue regeneration is the formation of new blood vessels. Neovascularization of wounds is made possible through angiogenesis or arteriogenesis. Angiogenesis can be induced by a heterogenous population of BM-SCs containing MSCs, MACs, and ECFCs, which are capable of forming a vascular endothelium [33]. As targets for IS drugs are also expressed on precursor cells, IS drugs could have a negative impact on the cell-based therapy.

However, the majority of studies on IS drugs are performed on patients without diabetes or using cells obtained from non-diabetic patients. DM patients have a decrease in the proliferation of vascular regenerative cells, as well as a reduction in their proangiogenic induction. This leads to several impairments that delay tissue healing [34,35,36]. Information is still limited regarding the impact of IS drugs on SCs and their vasculogenic potential in treating DPs. For this reason, we analyzed the effects of IS drugs on the characteristics and functional properties of BM-SCs derived from patients with DM. The mononuclear fraction of BM was cultured for 14–18 days, and the suspension of adherent and non-adherent cells was then phenotypically characterized by flow cytometry. BM-SCs that expressed CD90, CD105, and CD73, but not CD45, were characterized as MSCs. Cells expressing CD146 and not CD45 were classified as ECFCs. The populations expressing CD45 but not CD146 were classified as MACs.

To study the impact of IS drugs on the metabolic activity and cell death of BM-SCs, cells were cultured for 48 h in two ways: untreated or exposed to IS drugs. The results indicated that in concentrations of 50 µg/mL and higher, all the IS drugs significantly inhibited the metabolic activity of BM-SCs. Percentages of apoptotic and dead BM-SCs exposed to IS drugs were measured using flow cytometry. Every one of the IS drugs increased apoptosis of the BM-SCs. Sirolimus (5 µg/mL and higher) was the most effective stimulator of apoptosis. Bone marrow stem cell death increased in the presence of tacrolimus in concentrations of 0.005 µg/mL and higher. Other IS drugs also had a negative impact on the viability of BM-SCs. These observations suggest that IS drugs may decrease the survival rate of BM-SCs and potentially inhibit the therapeutic effects of autologous BM-SC therapy.

BM-SCs are able to modulate the responses of immune cells and downregulate inflammation in a wound [37,38], so we analyzed the effects of IS drugs on the expression of genes of immunoregulatory molecules in these cells. Tacrolimus downregulated the expression of genes for COX2, PD-L1, and IDO more effectively than the other IS drugs. Mycophenolate mofetil was the most potent inhibitor of TGF-β gene expression, and sirolimus significantly inhibited iNOS gene expression. These results suggest that IS drugs could decrease not only the immunoregulatory properties of BM-SCs but also their ability to modulate the function and activity of immune cells and inflammation.

The underlying mechanisms behind the effects of IS drugs on BM-SCs are still not fully understood. Tsuji et al. demonstrated that MSC derived from adipose tissue were more resistant to the cytotoxic effect of tacrolimus and anti-lymphocyte serum mediated through a complement system compared to MSC isolated from bone marrow [39]. They showed that those IS drugs had a negative impact on homing, survival, and key functional efficacy in vivo in rat models. Another study assessing murine heart models revealed that mycophenolate mofetil and cyclosporin A targeted different steps in the lymphocyte activation cascade—they blocked production of IL-2 and IFN-γ, disrupted the pro-inflammatory cytokine network controlled by T-helpers, and therefore prevented the activation of MSCs [40]. According to published studies and our own experiments, we assume that IS drugs influence BM-SCs on many different levels—they decrease viability and differentiation potential, block different cytokine pathways, and induce apoptosis and cell death.

Bone marrow precursor cells are used for the treatment of DFUs due to their ability to produce cytokines and growth factors. For this reason, we evaluated the impact of IS drugs on the production of HGF, CCL2, VEGF, IL-8, and IL-6. HGF is a growth factor that is involved in cell proliferation and angiogenesis [41], VEGF also contributes to angiogenesis and neovascularization [42], CCL2 and IL-8 are chemokines responsible for cell migration to the site of inflammation [43,44], IL-8 also contributes to angiogenesis [44], and IL-6 is a modulator of inflammation [45]. All of the IS drugs inhibited the ability of the BM-SCs to reproduce these cytokines and growth factors. Tacrolimus proved to be the most effective inhibitor of HGF, CCL2, IL-8, and IL-6 production. Sirolimus significantly decreased the ability of BM-SCs to produce VEGF. Altogether, these findings indicate that IS drugs may have an adverse effect on tissue regeneration.

In conclusion, our results show that IS drugs inhibit the characteristics and functional properties of BM-SCs from DPs and could, therefore, negatively impact ACT in DPs. These findings may have a direct impact on the indication of ACT in diabetic patients with the most severe stages of PAD—no-option chronic limb-threatening ischemia—who are seeking solid-organ transplantation. On the other hand, we did not observe a difference between patients treated with IS drugs with regard to the response to ACT in our clinical study [46]. The benefit of ACT in those patients needs to be carefully considered based on results of our current study, because the therapeutic potential of BM-SCs injected into the muscles of ischemic limbs would be decreased. One important question that remains is what the tissue concentrations of IS drugs are in those patients with severe PAD. Therefore, one of our future studies will assess the tissue levels of these drugs directly near foot ulcers using the microdialysis techniques.

## 5. Conclusions

To summarize, we have demonstrated that IS drugs negatively affect metabolic activity, cell death, the expression of genes of immunoregulatory molecules, and the ability of BM-SCs to produce cytokines and growth factors in patients with DFU. BM-SCs, which are involved in tissue healing, are used for cell-based therapy in DPs with DFUs. The healing effects of BM-SCs could become diminished in DPs who have undergone solid-organ transplantation and are receiving IS treatment. For this reason, the dose and type of IS medication should always be carefully considered and monitored.

## Figures and Tables

**Figure 1 biomedicines-11-01872-f001:**
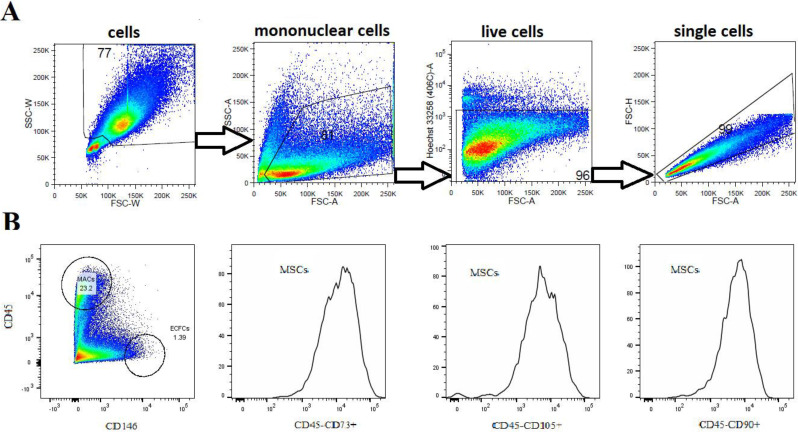
Characterization of BM-SC populations. (**A**) Representative dot plots show gating strategy of labeling live single-cell suspension analyzed by flow cytometry. (**B**) Representative histograms show that a single-cell suspension cultured for 14–18 days contained CD45^−^CD90^+^CD73^+^CD105^+^ cells (MSCs), and dot plot shows CD45^+^CD146^−^ cells (MACs and lymphocyte populations), CD45^−^CD146^+^ cells (ECFCs) and other cell populations.

**Figure 2 biomedicines-11-01872-f002:**
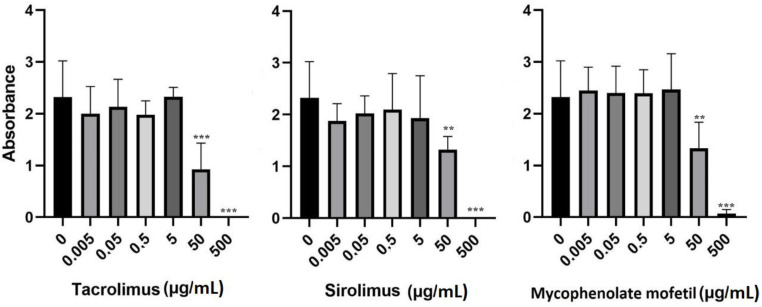
The effect of IS drugs on the metabolic activity of BM-SCs. Cells were cultured for 48 h in the presence of tacrolimus, sirolimus, or mycophenolate mofetil, and metabolic activity was measured by WST-1 assay. Each bar represents the mean + SD measured in triplicate from 6 patients. Values with asterisks are significantly different from untreated control “0” (** *p* < 0.01; *** *p* < 0.001).

**Figure 3 biomedicines-11-01872-f003:**
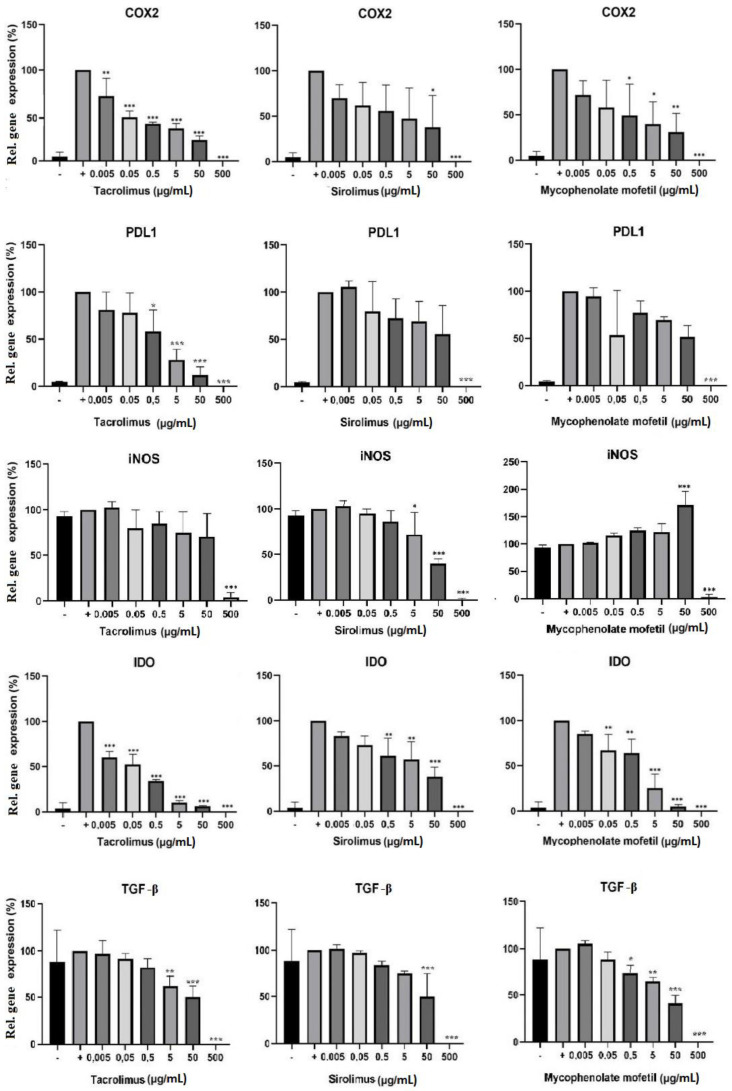
The effect of IS drugs on the expression of genes of immunoregulatory molecules in BM-SCs. Cells were cultured for 48 h in 3 different ways: (1) unstimulated (−), (2) stimulated with IFN-γ and TNF-α (+), or (3) stimulated with cytokines in the presence of sirolimus, tacrolimus, or mycophenolate mofetil. The expression of genes of immunoregulatory molecules IDO, iNOS, COX2, TGF-β, and PD-L1 was detected by RT-PCR. Each bar represents the mean + SD measured in triplicate from 6 patients. Values with asterisks are significantly different from stimulated control “+” (* *p* < 0.05; ** *p* < 0.01; *** *p* < 0.001).

**Figure 4 biomedicines-11-01872-f004:**
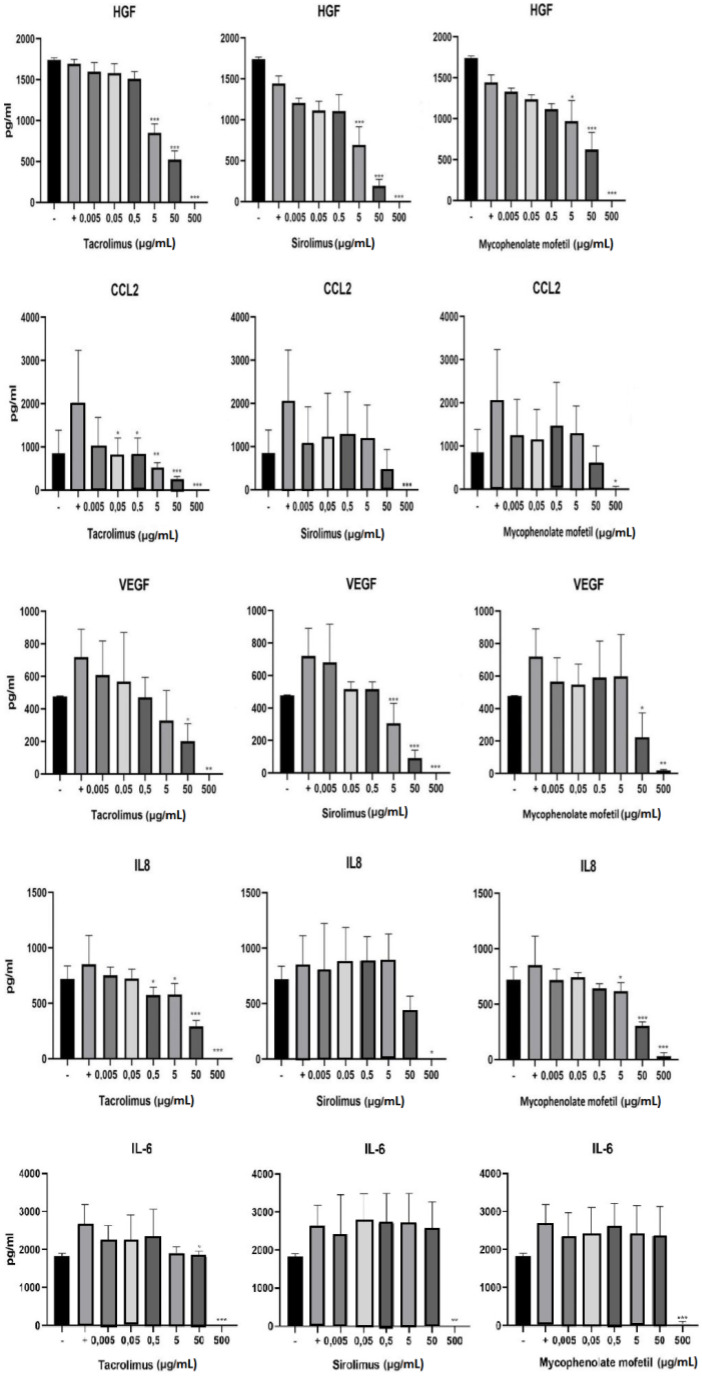
The impact of IS drugs on the production of cytokines and growth factors by BM-SCs. The cells were cultured in 3 different ways: (1) unstimulated (−), (2) stimulated with IFN-γ and TNF-α (+), or (3) stimulated with cytokines in the presence of various concentrations of tacrolimus, sirolimus, or mycophenolate mofetil. The production of cytokines by BM-SCs was determined in supernatants by ELISA. Each bar represents the mean + SD measured in triplicate from 6 patients. Values with asterisks are significantly different from stimulated control “+” (* *p* < 0.05; ** *p* < 0.01; *** *p* < 0.001).

**Figure 5 biomedicines-11-01872-f005:**
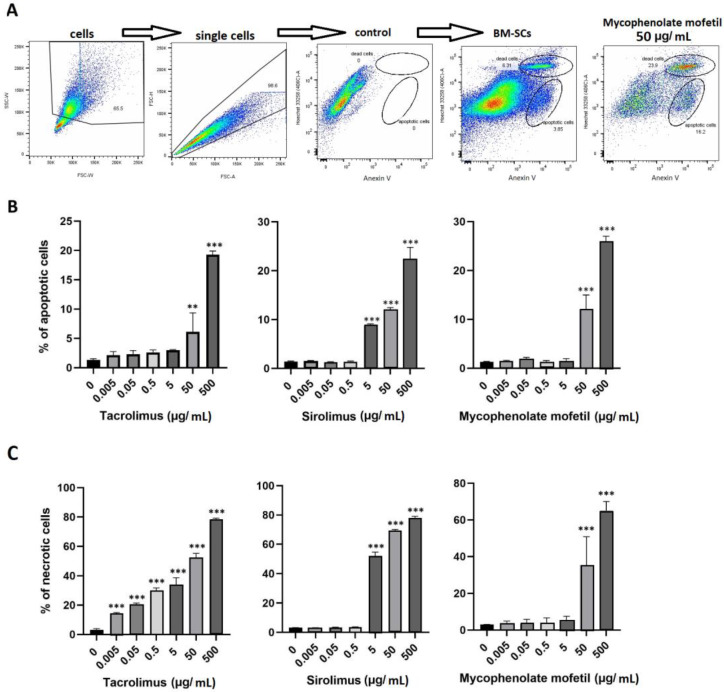
The effect of IS drugs on apoptosis (**B**) and cell death (**C**) of BM-SCs. The cells were cultured for 48 h untreated (0) or in the presence of different concentrations of tacrolimus, sirolimus, or mycophenolate mofetil, and the percentages of apoptotic and dead cells were measured by flow cytometry. Representative dot plots show the common gating strategy of labeling single-cell suspension, control (Anexin V^−^ Hoechst3358^−^), and both groups of cultured BM-SCs: untreated and exposed to mycophenolate mofetil (50 µg/mL) (**A**). Each bar represents the mean + SD measured in triplicate from 6 patients. Values with asterisks are significantly different from control “0” (** *p* < 0.01; *** *p* < 0.001).

**Table 1 biomedicines-11-01872-t001:** Primers used for amplification.

Gene	Sense Sequence 5′-3′	Antisense Sequence 5′-3′
GAPDH	GCCCAATACGACCAAATCC	AGCCACATCGCTCAGACAC
COX2	GCTGGCCCTCGCTTATGA	GCTCAAACATGATGTTTGCATTC
PD-L1	GGTGAGGATGGTTCTACACAG	GAGAACTGCATGAGGTTGC
iNOS	GCTCTACACCTCCAATGTGACC	CTGCCGAGATTTGAGCCTCATG
IDO	CATCTGCAAATCGTGACTAAG	CAGTCGACACATTAACCTTCCTTC
TGF-β	TATCGACATGGAGCTGGTGAAG	CAGCTTGGACAGGATCTGGC

## Data Availability

Data of the patients are unavailable due to privacy and ethical restrictions.

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
