# Peer review of "The Effects of Immunosuppressive Drugs on the Characteristics and Functional Properties of Bone Marrow-Derived Stem Cells Isolated from Patients with Diabetes Mellitus and Peripheral Arterial Disease"

_biomedicines, 2023, doi:10.3390/biomedicines11071872_

Round 1
Reviewer 1 Report
In this manuscript, the authors investigated the impact of immunosuppressive drugs on bone marrow-derived stem cells. They reported that three immunosuppressive drugs led to reduced metabolic activity, altered immunoregulatory markers, and decreased cell viability. However, it is important to note that these findings are based solely on in vitro assays, which may limit the strength of the conclusions drawn. Moreover, the study lacks a comprehensive mechanistic investigation and lacks a proper internal control. Consequently, more data are necessary to substantiate the conclusions drawn. Therefore, I have several major comments regarding the manuscript:
1. In Figure 1B, the population identified as myeloid angiogenic cells is labeled as CD45+CD146-. However, it should be acknowledged that aside from myeloid cells, this population may also contain various lymphocytes. Therefore, it is recommended that the authors rename these populations accordingly.
2. Regarding the in vitro assay, the authors mentioned that the immunosuppressive drugs tacrolimus, sirolimus, and mycophenolate mofetil were used at concentrations ranging from 0.005 to 500 μg/mL. It would be valuable to clarify the solvent used to dissolve and stock these drugs. Were DMSO or water used? Additionally, it is important to consider that higher amounts of DMSO can impact cell viability and metabolic activity. To mitigate this concern, it is suggested to include a DMSO-only group (or a solution-only control group) to assess any potential side effects arising from high concentrations of the solvent.
3. The authors exclusively examined the effects of immunosuppressive drugs on bone marrow-derived stem cells. However, it would be beneficial to explore the impact of these drugs on other cell types, such as lymphocytes, granulocytes, monocytes, and hematopoietic stem cells (HSCs), to gain a more comprehensive understanding of their effects.
4. In Figure 3, mRNA levels were assessed. It would be valuable to complement these findings by presenting protein-level data as well, through techniques such as Western blotting or flow cytometry.
5. The manuscript would greatly benefit from an exploration of the underlying mechanisms behind the effects of immunosuppressive drugs on bone marrow-derived stem cells. Elucidating these mechanisms would enhance the overall understanding of the study's findings.
6. Finally, it is important for the authors to discuss the potential translational value of their research and provide insights into potential therapeutic solutions based on their findings.
Author Response
Dear reviewer,
thank you a lot for many important comments and suggestions that improved our manuscript. We changed and added new information adn marked all in red font.
Please see the attachment for the point-by-point response for your comments.
Best regards,
Michal Dubsky, corresponding author

Reviewer 2 Report
This study determines the effects of immunosuppressive drugs on the characteristics of bone marrow-derived cells. Cells were isolated from the bone marrow of diabetic patients, cultured for 2+ weeks and then characterized by flow cytometry. The metabolic, gene expression, cytokine secretion and survival effects of immunosuppressive drugs was determined on cultured cells. The authors characterized the cells as mesenchymal SCs (defined as CD45-CD73+CD90+CD105+), myeloid angiogenic cells (CD45+CD146-) and endothelial colony-forming cells (CD45-CD146+). Overall, results showed that immunosuppressive drugs had a negative impact on their cultured cells in a dose-dependent manner. As these cells are used to treat ischemic diabetic foot ulcers, the authors suggest this likely influences therapeutic potential.
Overall, this is a very appropriate addition to the special issue on "Peripheral Artery Disease and Diabetic Foot Ulcer: From Bench to Clinic". The authors do need to be more specific in their use of the term "stem cells (SC)", however. This term is used far too loosely in this manuscript.
Some issues, but overall this can be fixed with some moderate editing.
Author Response
Dear reviewer, thank you a lot for important comments and suggestions that improved our manuscript. We did a complete check of the whole manuscript of the native speaker and he dida lot of English changes. All changes are marked in manuscript i red font.
Please see our response in the attached file.
Best regards,
Michal Dubsky, corresponding author.

Round 2
Reviewer 1 Report
No further comment